# Retrieval and Mapping of Soil Organic Carbon Using Sentinel-2A Spectral Images from Bare Cropland in Autumn

**Ke Wang** , **Yanbing Qi \*** , **Wenjing Guo, Jielin Zhang and Qingrui Chang**

College of Natural Resources and Environment, Northwest A&F University, Yangling 712100, China;
wk@nwafu.edu.cn (K.W.); guowenjing@nwafu.edu.cn (W.G.); zhangjielin1027@nwafu.edu.cn (J.Z.);
changqr@nwsuaf.edu.cn (Q.C.)

\* Correspondence: ybqi@nwsuaf.edu.cn; Tel.: +86-298-708-2912; Fax: +86-298-708-0055

**Abstract:** Soil is the largest carbon reservoir on the terrestrial surface. Soil organic carbon (SOC) not only regulates global climate change, but also indicates soil fertility level in croplands. SOC prediction based on remote sensing images has generated great interest in the research field of digital soil mapping. The short revisiting time and wide spectral bands available from Sentinel-2A (S2A) remote sensing data can provide a useful data resource for soil property prediction. However, dense soil surface coverage reduces the direct relationship between soil properties and S2A spectral reflectance such that it is difficult to achieve a successful SOC prediction model. Observations of bare cropland in autumn provide the possibility to establish accurate SOC retrieval models using the S2A super-spectral reflectance. Therefore, in this study, we collected 225 topsoil samples from bare cropland in autumn and measured the SOC content. We also obtained S2A spectral images of the western Guanzhong Plain, China. We established four SOC prediction models, including random forest (RF), support vector machine (SVM), partial least-squares regression (PLSR), and artificial neural network (ANN) based on 15 variables retrieved from the S2A images, and compared the prediction accuracy using RMSE (root mean square error), $R^2$ (coefficient of determination), and RPD (ratio of performance to deviation). Based on the optimal model, the spatial distribution of SOC was mapped and analyzed. The results indicated that the inversion model with the RF algorithm achieved the highest accuracy, with an $R^2$ of 0.8581, RPD of 2.1313, and RMSE of 1.07. The variables retrieved from the shortwave infrared (SWIR) bands (B11 and B12) usually had higher variable importance, except for the ANN model. SOC content mapped with the RF model gradually decreased with increasing distance from the Wei river, and values were higher in the west than in the east. These results matched the SOC distribution based on measurements at the sample sites. This research provides evidence that soil properties such as SOC can be retrieved and spatially mapped based on S2A images that are obtained from bare cropland in autumn.

**Keywords:** soil organic carbon; sentinel-2A; random forest; seasonal bare cropland; variable importance

## 1. Introduction

In recent decades, increased carbon emission and its potential effects on ecological systems has aroused great concerns globally. The closely related regional carbon storage and cycles have also become an intensively researched topic in macroecology [1]. Soil carbon banks, with a soil carbon reserve of about 1500 Pg C, are one of the key components of the global carbon cycle [2], and are more than twice the atmospheric carbon bank reserves and three times the terrestrial biocarbon reserves [3]. Soil carbon includes organic and inorganic fractions. Inorganic carbon is distributed in the deep soil layers with a longer cycling period, while organic carbon is distributed in the top soil layers (within 1 m of the soil surface) where carbon cycles directly among soil, air, and plants [4]. Due to the huge reserves of soil organic carbon (SOC), small variations of SOC in the soil by

accumulation (forestation) or decomposition (logging, incineration) can cause significant fluctuations in $CO_2$ concentrations in the atmosphere [5,6]. Land use is one of the factors affecting SOC storage and variation. Especially in the case of farmland, which accounts for 10.7% of the world land area, long-time crop production practices, such as fertilization, tillage, and irrigation, frequently change SOC status. Meanwhile, SOC contributes to plant growth through its effect on the soil physical, chemical, and biological properties, and has been considered as the most important soil quality indicator. Therefore, detecting SOC status in farmland and mapping its spatial variation is vital for monitoring soil quality and producing environmental benefits [7].

The soil is an exceptionally variable, heterogeneous environment, both spatially and temporally. In order to acquire the spatial distribution of SOC, soil sampling and laboratory determination are required, both of which are time-consuming and expensive [8,9]. With the rapid development of remote sensing in recent years and its application to soil, retrieval of soil properties based on reflectance spectroscopy has become an alternative way to monitor field soil [10]. However, surface vegetation can block most of the soil spectrum information obtainable from satellites. It is difficult for satellite sensors to directly detect the reflectance characteristics of soil during crop growing seasons, and this interference by vegetation significantly reduces remote sensing's prediction accuracy of soil properties. Farmland does, however, have a crop-free period in which the soil is bare, and this period provides the possibility of enhancing soil property prediction accuracy from satellite imagery of exposed soils [11]. The visible-near-infrared and shortwave infrared hyperspectral images can be used to quantitatively provide soil property information by intensively and continuously collecting spectral information of bare soil in certain optical domains. Castaldi et al. [12] and Laamrani et al. [13] obtained reliable soil property prediction results from hyperspectral images. However, few hyperspectral sensors are currently available as both spaceborne and airborne instruments. In recent years, researchers have given great attention to the correlation between airborne multispectral remote sensing data and soil properties, as well as to the applicability of multispectral remote sensing data of exposed soils as captured by the SPOT (Systeme Probatoire d'Observation de la Terre) constellation of satellites [14,15]. It is possible to quantitatively estimate soil attributes using multispectral or hyperspectral data as long as high-quality spectral data in certain sensitive bands are obtained. However, prediction accuracy can yet be improved. Fortunately, in 2015 the European Space Agency successfully launched the Sentinel-2A (S2A) satellite that was equipped with high-quality multispectral (7–20 band) image sensors. Access to the images was provided free of charge, and undoubtedly provided the opportunity for multispectral data to be used to obtain a wide range of quantitative soil attribute information.

The S2A images have been successfully used in various fields since their date of availability. Because the S2A has four bands in the red edge area of vegetation, scholars first conducted a great amount of research on vegetation monitoring and classification, in which they have made some progress. For example, He et al. [16] assimilated S2A leaf area index time series images into an ecological model to predict cotton yield, with results showing that 85% of the variation in yield was explained. They concluded that these images could be used as a data source for accurate estimation of cotton yield. Chen et al. [17] obtained an enhanced flower index (EBI) closely related to vegetation flower phenology based on S2A images. It was expected that EBI obtained by satellite would track the flowering information, and thereby improve our understanding and prediction of the response of flowers and pollination to weather and final yield. Yun et al. [18] used S2A multi-band spectral characteristics, normalized difference vegetation index (NDVI), ratio vegetation index (RVI), difference vegetation index, normalized difference water index (NDWI), and other index characteristics, as well as contrast, correlation, energy, mean value, entropy, and other texture characteristics to study the southeast area of the Mu River Basin in the southern central peninsula under the framework of a random forest model. The accuracy of land cover classification was 87.53% and the kappa coefficient was 0.8461, and these results were better than the original random forest method. These

results will undoubtedly contribute significantly to the further development of precision agriculture [19]. In addition, S2A datasets have made corresponding contributions to the study of glaciers [20], volcanoes [21], and reservoir water purity assessment [9]. However, S2A research regarding applications to topsoil properties (including CEC, pH, SOC, calcium carbonate, texture, etc.) has only appeared in recent years [22,23]. Gholizadeh et al. [24] assessed the potential of using S2A data to study soil texture and organic carbon monitoring and mapping through quantitative regional prediction and mapping of soil parameters in central Europe. They compared the accuracy results with those obtained from airborne hyperspectral data and laboratory spectral data. Results derived from S2A were slightly lower than those obtained from laboratory spectra and airborne images, but the decrease in accuracy may be offset by the extensive geographic coverage and greater frequency of satellite observations. In this case, some researchers also obtained similar results [25–27]. However, due to the strong regional variability of surface soil properties, the prediction model obtained at the small local scale was not extremely versatile, and could not be extended to other regions. Information mining of surface soil properties based on S2A data still needs further research.

Previous studies have tried a variety of SOC hyperspectral prediction models to obtain the best prediction accuracy. Machine learning is an effective empirical approach applied in many fields of earth science to produce successful soil property prediction models [28,29]. These machine learning methods include random forest (RF), partial least squares regression (PLSR), support vector machine (SVM), and artificial neural network (ANN). RF is an ensemble algorithm with great potential to remove over-fitting, and it gained great popularity in classification and regression [30,31]. ANN models learn from training datasets, and mimic some intelligent behaviors of the human brain, widely applying a large number of learn-from-data applications in such areas as water resources and hydrology [32–34]. Two other data-driven models (SVM and PLSR) excel at approximating multivariate non-linear relationships among the target variables by seeking a new coordinate system [35,36]. Significant differences have been observed in the best estimation models for different soil types as well as land uses, and further attempts need to be explored to achieve high prediction accuracy adapted to local conditions.

Currently, hyperspectral SOC prediction research has achieved abundant progress, but no unified models in existing studies have produced agreement, and great variation in model estimation accuracy has been detected. Therefore, a model established in one region has been impossible to apply to another region, and the prediction results have not been comparable. Furthermore, most SOC spectral predictions have been focused on areas with vegetation growing over the soil surface (e.g., forest and cropland), and there have been few reports regarding bare cropland topsoil. The objective of this study was to assess the capacity of S2A images to predict SOC content and to map spatial distribution of SOC. This study focuses on the western Guanzhong Plain in Shaanxi Province, China, where bare farmland was investigated in autumn. Four machine learning algorithms (RF, PLSR, SVM, and ANN) were used to link the local surface spectral data with field observations without considering influence of soil moisture on SOC retrieval, as well as 15 variables retrieved from S2A images in order to predict SOC content of the first 30 cm based on reflected spectral values of bare farmland topsoil.

## 2. Materials

### 2.1. Study Areas

We selected the western Guanzhong Plain for our SOC field study area (Figure 1). This is a dominant crop production area in Shaanxi province, China, where bare arable land could be detected in autumn. The western Guanzhong Plain covers Fufeng county, Qishan county, Fengxiang county, and the eastern part of Chencang district, encompassing latitudes 34°13′ to 34°47′ N and longitudes 107°10′ to 108°4′ E, and covering a total area of 2926 km². The study area was adjacent to the Loess Plateau in the north and the Qinling Mountains (running east to west) in the south. The landforms observed in this region can

be divided into alluvial flat terrace along the Wei river, low tableland in the transitional area of the Loess Plateau, and the Wei river valley. Landform gently changes from high in the west to low in the east, with the altitude ranging from 0 to 1663 m. This area is dominated by a temperate, semi-arid continental monsoon climate. The average annual precipitation ranges from 550 to 750 mm, 70% of which falls between June and September. Sunshine duration is 2300 h with a sunshine rate of 62%. The mean annual air temperature ranges from 12 to 14 °C, and accumulated temperature (>10 °C) is about 4500 °C.

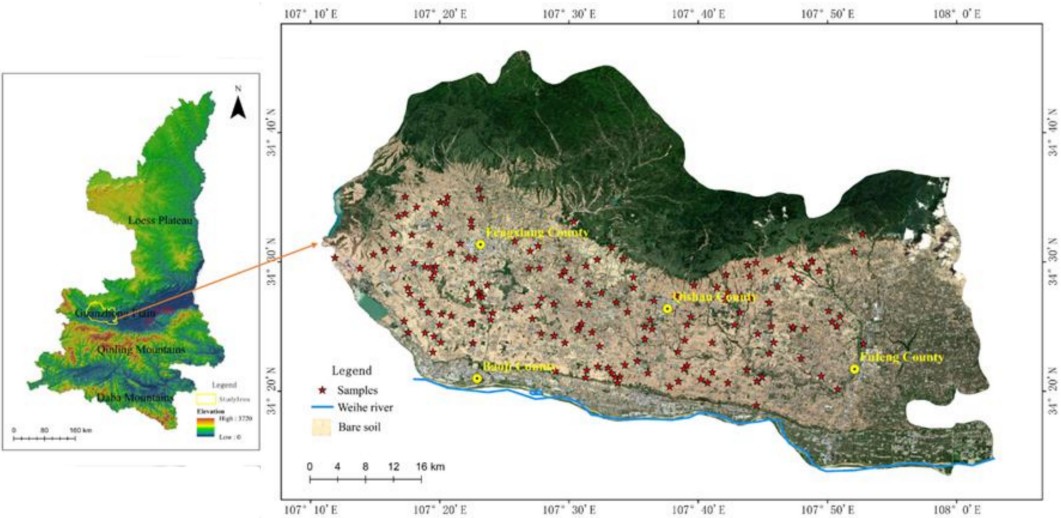

**Figure 1.** Location of autumnal bare farmland sample sites in the western Guanzhong Plain, China (the images containing bare farmland were obtained from Landsat 8 OLI on 12 August and 15, 2019).

The selected study area includes three of the 23 dominant crop production counties in Shaanxi province because of the appropriate climate conditions, fertile soils, and flat landform. The dominant soil groups are Hapli-Ustic Argosols, Earth-cumuli-Orthic Anthrosols, and Ochri-Aquic Cambosols (the corresponding soil groups were Hapudalfs, Hapustalfs, and Haplustepts, respectively, based on the soil taxonomy in the Wei River Valley [37]). Most of the land in the study area is used for crop and fruit production.

### 2.2. Soil Sampling and S2A data

The climate and soil conditions in the research area are suitable for the traditional double cropping system of winter wheat followed by autumn maize. However, we found that most of the arable land was not used for crop production after wheat was harvested in June until wheat was planted in October (as shown in Figure 1). We investigated the research area in August 2019. Based on the field investigation and remote sensing images, 225 sample sites were randomly and relatively homogeneously selected in the autumn bare arable land area during the period of 11–20 August 2019, and this period was consistent with the satellite passing time of S2A. Latitude and longitude of each sample site were recorded using a hand-held global positioning system (GPS). Topsoil samples (0–15 cm) were collected using an auger with 10 cm diameter at each sampling site, and included five topsoil subsamples within an area of 50 m². Following collection, the five subsamples were combined and mixed to form a single 500 g final sample. The soil samples were air-dried and sieved through a 2-mm sieve prior to analysis. SOC was determined by the dichromate-wet combustion method [38]. These data were then used to build four retrieval models and a SOC map (Figure 2).

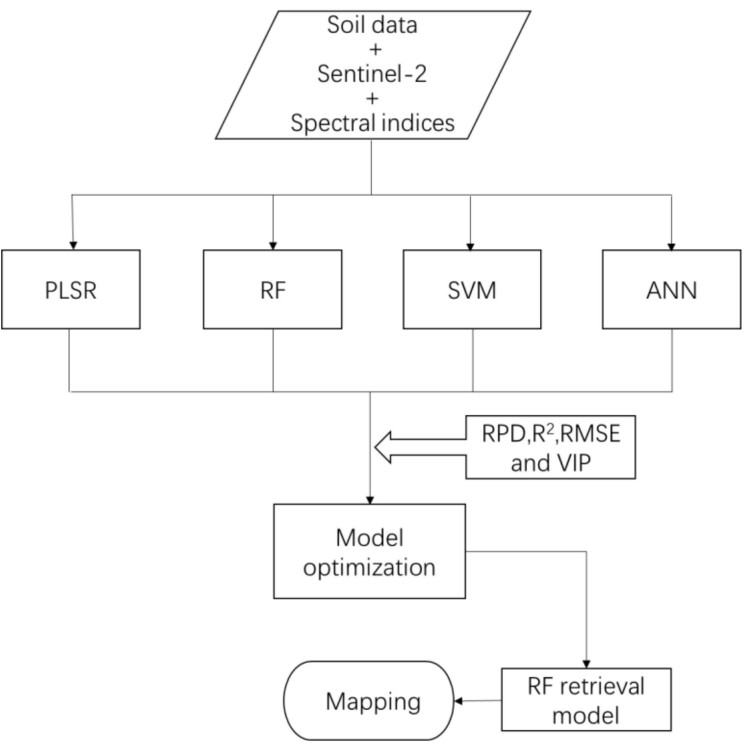

**Figure 2.** Flowchart showing the soil organic carbon (SOC) prediction method. PLSR = partial least squares regression; RF = random forest; SVM = support vector machine; ANN = artificial neural network; RPD = relative percent difference; $R^2$ = coefficient of determination; RMSE = root mean square error; VIP = variance importance in projection.

　　　Retrieval of SOC for bare arable land was based on S2A satellite images. The S2A satellite provides multispectral reflectance remote sensing images that comprised 13 successive spectral bands within the range of 443–2190 nm [39]. The S2A has a 5-day revisit cycle (Table 1). Different bands had one of three spatial resolutions: 10-m resolution was assigned to B2, B3, B4, and B8 with central wavelengths of 490 nm, 560 nm, 665 nm, and 842 nm, respectively; 20-m resolution was assigned to B5, B6, B7, B8A, B11, and B12 with central wavelengths of 705 nm, 740 nm, 783 nm, 865 nm, 1610 nm, and 2190 nm, respectively; and 60-m resolution was assigned to the B1, B9, and B10 bands that were used as the atmospheric correction channels with central wavelengths of 443 nm, 945 nm, and 1375 nm, respectively. The autumnal bare farmland was easily detected in August based on our field investigation. Therefore, three S2A multispectral images with cloud fractions less than 20%, and no significant cloud cover in the study area were downloaded for data collected on August 12 and 15, 2019, from the Copernicus open access hub (Table 1) (https://scihub.copernicus.eu/dhus/#/home (accessed on 2 February 2021)). The atmospheric, terrain, and cirrus corrections for the obtained S2A images were performed by the Sen2cor processor [40,41]. The top of the atmosphere reflectance was converted into the bottom of atmosphere reflectance based on a look-up table algorithm [42]. Due to differences in ground sampling distances among different bands in the image, the nearest neighbor resampling method was used to unify the spatial resolution of the atmospherically corrected images to 20 m to comprehensively reflect the level of detail of S2A data. Doing so not only preserves the input image pixel values, but also greatly improves computational efficiency. Nine of the 13 available spectral bands (B2, B3, B4, B5, B6, B7, B8A, B11, and B12) were selected as indices for retrieving SOC (shown in bold in Table 2). Among the spectral bands, B8A (855–875 nm) characterizes soil spectral absorption characteristics and has higher spatial resolution [22], so B8 (789–900 nm) is not required to participate in the model. Likewise, B1, which is used to monitor aerosol in coastal water and the atmosphere,

B9, used to characterize water vapor, and B10, used to characterize cirrus clouds, were not selected.

**Table 1.** Main characteristics of the studied scenes acquired over the study area, the western Guanzhong Plain, China.

| Imaging Date | Sensor | Output Resolution (m) | Time of Acquisition (U.T GMT) | Solar Azimuth (°) | Solar Elevation (°) |
|---|---|---|---|---|---|
| 12 August 2019 | S2A | 20 | 03:25:41 | 134.5 | 64.05 |
| 15 August 2019 | S2A | 20 | 03:35:41 | 138.5 | 64.17 |
| 15 August 2019 | S2A | 20 | 03:35:41 | 140.5 | 64.76 |

**Table 2.** Characteristics of the multispectral instrument aboard the Sentinel-2A satellite. Bands shown in bold font represent those selected for retrieving soil organic carbon content.

| Sentinel-2 Bands | Spectral Position (nm) | Central Wavelength (nm) | Resolution (m) | SNR (at Lref) [1] |
|---|---|---|---|---|
| B1-Coastal aerosol | 421–557 | 443 | 60 | - |
| **B2**-Blue | 458–523 | 490 | 10 | 154 |
| **B3**-Green | 543–578 | 560 | 10 | 168 |
| **B4**-Red | 650–680 | 665 | 10 | 142 |
| **B5**-Vegetation Red Edge | 698–713 | 705 | 20 | 117 |
| **B6**-Vegetation Red Edge | 733–748 | 740 | 20 | 89 |
| **B7**-Vegetation Red Edge | 773–793 | 783 | 20 | 105 |
| B8-NIR | 789–900 | 842 | 10 | 174 |
| **B8A**-Vegetation Red Edge | 855–875 | 865 | 20 | 72 |
| B9-Water vapor | 931–958 | 945 | 60 | - |
| B10-SWIR-Cirrus | 1338–1414 | 1375 | 60 | - |
| **B11**-SWIR | 1565–1655 | 1610 | 20 | 100 |
| **B12**-SWIR | 2100–2280 | 2190 | 20 | 100 |

[1] SNR, signal-to-noise ratio and the MSI (Multispectral Imager) instrument specifications for SNR are generally at Lref.

An inspection of Figure 1 shows the dominant land coverage to be arable land with scattered areas occupied by buildings and water bodies. To accurately acquire the bare soil pixels in the S2A data, the water bodies and buildings were identified and masked. After the arable land pixels with NDVI values greater than 0.2 were masked, the rest of the arable land was identified as bare land, accounting for about 26% of the study area. Additionally, even though the S2A spectral reflectance is the result of many soil factor especially soil moisture and soil texture, SOC has a significant negative correlation with spectral reflectance, i.e., a lower SOC concentration corresponds to bare soil pixels with high reflectance and a higher SOC concentration corresponds to bare soil pixels with low reflectance (Figure 3). These curves corresponded to maximum and minimum, as well as to two random values of measured SOC between the maximum and minimum values.

The spectral reflectance remote sensing data provide a comprehensive identification of various ground objects including vegetation, water, and soil [43,44]. The characteristics of S2A spectral reflectance for bare soils significantly reflected several soil physical and chemical properties, such as soil water, texture, iron, organic carbon, $CaCO_3$, and salinity. The spectral signature of certain soil properties is characterized by the signature's general shape, reflectance intensity, and specific absorption bands. For soil properties, SOC concentration is a synthetic reflectance of soil conditions such as plant growth, water, and soil color. Therefore, in addition to the selected nine S2A spectral bands, several

spectral indices were integrated into the retrieving models that included vegetation indices (that are sensitive to biomass, that is, sensitive to organic carbon), water indices (that are sensitive to soil moisture), and brightness-related indices (that are sensitive to both organic carbon content and soil texture) [24]. The selected specific spectral indices included NDVI [45], Transformed Vegetation Index (TVI) [46], Enhanced Vegetation Index (EVI) [47], Green Normalized Difference Vegetation Index (GNDVI) [48], Green-Red Vegetation Index (GRVI) [49], Moisture Stress Index (MSI) [50], Soil-Adjusted Vegetation Index (SAVI) [51], Soil-Adjusted Total Vegetation Index (SATVI) [52], the Second Modified Soil Adjusted Vegetation Index (MSAVI2) [53], Brightness Index (BI) [54], the Second Brightness Index (BI2) [54], Redness Index (RI) [55], Color Index (CI) [56], and vegetation (V) [56]. The specific calculation formulae are shown in Table 3. Six of the 14 spectral bands were selected as the indices for SOC retrieving (shown in bold in Table 3), filtered by the weighted average of variable importance in each retrieval model (Figure 4) and environment in study area. The threshold we chose for their exclusion was 0.023, in which the nine bands mentioned above as well as SATVI, BI2, and TVI were chosen. Nevertheless, to avoid a small amount of vegetation cover on the soil, several vegetation indices sensitive to low vegetation cover and soil were selected, including NDVI, SAVI, and EVI.

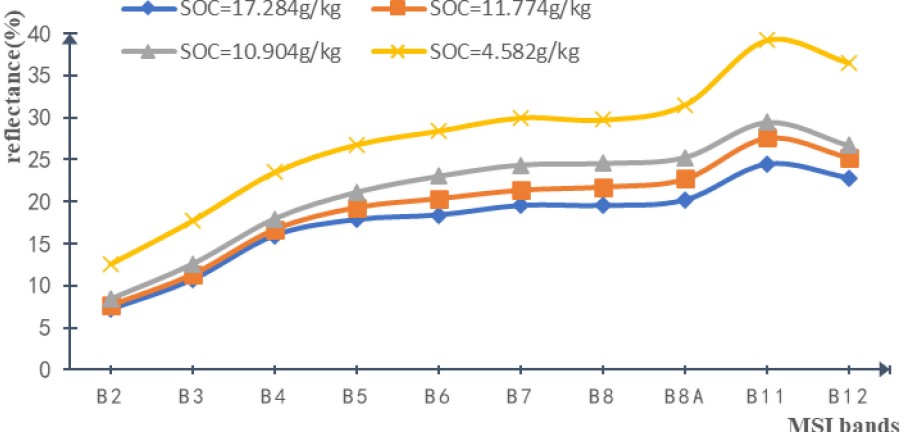

**Figure 3.** Soil spectral reflectance corresponding to maximum and minimum values of measured soil organic carbon (SOC), as well as to two random values of measured SOC.

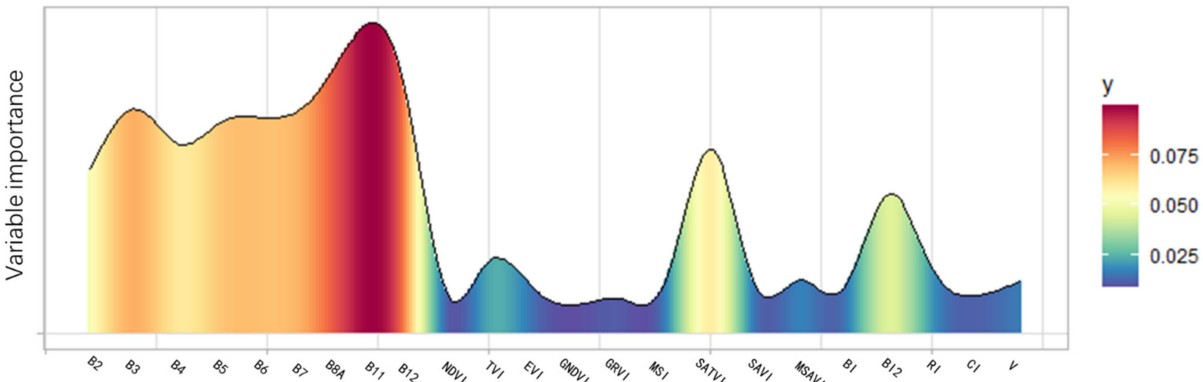

**Figure 4.** Distribution of variance importance in projection values for bands and all spectral indices.

**Table 3.** Definition and calculation of the selected specific spectral indices based on the Sentinel-2A image reflectance data (see index abbreviation definitions in the text).

| Index | Definition | Calculation Based on S2A Image Bands |
|---|---|---|
| **NDVI** | $\frac{NIR-Red}{NIR+Red}$ | $\frac{B8-B4}{B8+B4}$ |
| **TVI** | $\left(\frac{\rho NIR-\rho Red}{\rho NIR+\rho Red}+0.5\right)^{1/2}\times100$ | $\left(\frac{B8-B4}{B8+B4}+0.5\right)^{1/2}100$ |
| **EVI** | $2.5\frac{\rho NIR-\rho Red}{\rho NIR+6\times\rho Red-7.5\times\rho Blue+1}$ | $2.5\frac{B8-B4}{B8+6B4-7.5B2+1}$ |
| GNDVI | $\frac{\rho NIR-\rho Green}{\rho NIR+\rho Green}$ | $\frac{B8-B3}{B8+B3}$ |
| GRVI | $\frac{\rho Green-\rho Red}{\rho Green+\rho Red}$ | $\frac{B3-B4}{B3+B4}$ |
| MSI | $\frac{SWIR1}{NIR}$ | $\frac{B11}{B8}$ |
| **SATVI** | $\frac{\rho SWIR1-\rho Red}{\rho SWIR1+\rho Red+1}\times2-\frac{\rho SWIR2}{2}$ | $\frac{B11-B4}{B11+B4+1}2-\frac{B12}{2}$ |
| **SAVI** | $\frac{(\rho NIR-\rho Red)\times1.5}{\rho NIR+\rho Red+0.5}$ | $\frac{(B8-B4)\times1.5}{B8+B4+0.5}$ |
| MSAVI2 | $\frac{2\times\rho NIR+1-\sqrt{(2\times\rho NIR+1)^2-8\times(\rho NIR-\rho Red)}}{2}$ | $\frac{2\times B8+1-\sqrt{(2\times B8+1)^2-8\times(B8-B4)}}{2}$ |
| BI | $\sqrt{(\rho Red\times\rho Red)+(\rho Green\times\rho Green)}/2$ | $\sqrt{(B4\times B4)+(B3\times B3)}/2$ |
| **BI2** | $\frac{\sqrt{(\rho Red\times\rho Red)+(\rho Green\times\rho Green)+(\rho NIR\times\rho NIIR)}}{2}$ | $\frac{\sqrt{(B4\times B4)+(B3\times B3)+(B8\times B8)}}{3}$ |
| RI | $\frac{\rho Red\times\rho Red}{\rho Green\times\rho Green\times\rho Green}$ | $\frac{B4\times B4}{B3\times B3\times B3}$ |
| CI | $\frac{\rho Red-\rho Green}{\rho Red+\rho Green}$ | $\frac{B4-B3}{B4+B3}$ |
| V | $\frac{\rho NIR}{\rho Red}$ | $\frac{B8}{B4}$ |

## 3. Methods

### 3.1. Models for SOC Prediction

We tested four different multivariate models for retrieving SOC concentration from spectral data: PLSR, SVM, RF, and ANN. The variable importance was calculated for each regression method. Prediction accuracy was compared among the models.

The PLSR method is a mathematical optimization technology [57]. The method constructs a multivariate regression model by compressing uncorrelated independent variables. That is, the independent variable and dependent variable are projected into a new coordinate system so that it is relatively easy to construct a linear correlation between the two datasets. In this coordinate system, the multi-linear model is constructed, and it is usually used in situations where the variable dimension is equal to or less than the observation value and the multiple correlation between variables. For example, a spectral band and optical transmission model can be integrated with principal component analysis and typical correlation analysis methods to extract more abundant and deeper information from the original data. In this study, the matrix of the independent variable (spectral band spectral index) and the dependent variable (measured value of SOC) is taken into account in the selection of PLSR components, the number of which was set by selecting what number provided the lowest root mean square error (RMSE) for 10-fold cross validation. Finally, the variance importance in projection (VIP) values were calculated to estimate the importance of each independent variable in the PLSR model [25,58], taking into account the amount of explained variance for each component of the model.

The SVM regression is an algorithm based on statistical learning theory that belongs to supervised learning [24,59]. It is mainly applied to situations where the sample size is not massive and the feature dimension is far less than the sample number. Based on the selected kernel function, the techniques are able to approximate nonlinear relationships between multidimensional spaces and to derive a linear hyperplane as a decision function for nonlinear problems. In this paper, the soil spectral model was constructed based on the radial basis kernel, and the best parameters for the model were determined by the grid search method.

The RF composed of decision trees is one of an ensemble of learning algorithms, and is widely used for classification and regression problems [60]. Similar to SVM, it is also a supervised learning algorithm that combines multiple regression trees by resampling of the

training dataset, and then constructs a "forest." In this case, the user inputs the dependent data (training dataset) and independent data (remote sensing data), and then lets each tree in the forest judge and estimate separately, with each tree getting its own value. Then, the results of all decision trees are averaged as the outputs of the model. Therefore, to a great extent, this method avoids over-fitting issues. In addition, the relative variable importance (RVI) in the model can be calculated, and is used to evaluate the contribution of each independent variable to the model. The mean square error before and after permuting each predictor is calculated, after which the mean difference of all trees is normalized to the standard deviation for each variable. The RVI can be obtained by dividing these values by their sum [25].

The ANN is an unsupervised learning algorithm that is mainly used for classification and regression. It refers to clustering unlabeled data, determining the category of data, and predicting continuous values [61]. The ANN regression algorithm mainly includes three layers: input layer, hidden layer, and output layer. The hidden layer performs a weighted linear combination of multiple independent variables. Before output, a nonlinear function is used to modify the combination result to reduce the influence of extreme input values. In this process, the variable weight (that is, the importance of the variable) needs to be restricted to prevent it from being too large. The parameter that limits the weight is called the attenuation parameter, and is usually set to 0.1. Different random starting points are selected to train for many times, and the VIP of each variable is obtained by averaging several results.

For the four regression models, the ratio of training datasets to validation dataset was 3:1. Prediction accuracy of the models was evaluated by *RMSE* (Equation (1)), coefficient of determination ($R^2$), and relative percent difference (*RPD*, Equation (2)). After discretization, a randomly sampled spectral dataset of bare land area in the study area was input into the best performing model to retrieve the SOC content, after which the SOC was mapped. In the model classification system of Chang et al. [62] when the *RPD* is less than 1.0, the prediction ability of the model is poor and is not recommended; when the *RPD* is between 1.0 and 1.4, the prediction ability is only enough to detect high and low values; when the *RPD* is between 1.4 and 2.0, prediction ability is fair; when the *RPD* is between 2.0 and 2.5, the model has strong prediction ability and quantitative inversion ability; and *RPD* values greater than 2.5 indicate excellent prediction ability. The construction and evaluation of the models were completed in the RStudio (https://rstudio.com/ (accessed on 2 February 2021)).

$$RMSE = \sqrt{\frac{\sum_{i=1}^{n}(y_0 - y_p)^2}{n}} \tag{1}$$

$$RPD = \frac{std}{RMSE} \tag{2}$$

where $y_0$ are the observed SOC values, $y_p$ are the SOC values predicted by the model, *std* is the standard deviation of the observed SOC values, and n is the number of samples.

### 3.2. Variograms of Predicted SOC

The SOC distribution map was obtained by applying the four predictive regression models to all bare soil fields. The predicted values of SOC content based on spectral image values were modeled by a geostatistical method. The spatial structure could be described according to the variogram related to spatial dependence or the semi variance between samples [63]:

$$\gamma(h) = \frac{1}{2n}\sum_{i=1}^{n}\{p(x_i) - p(x_i + h)\}^2 \tag{3}$$

where $\gamma(h)$ is the average semi-variance of SOC, *n* is the number of pairs of sampling points, *p* is the SOC value, *x* is the coordinate of the point, and *h* is the distance between pairs or the lag value.

The semi-variance was calculated from a series of discrete distances (*h*) between the SOC point pairs (Equation (3)). As one of the theoretical functions, the variance model was obtained by fitting the variance to analyze the variation of SOC under any *h* [64]. All of the geostatistical analyses were performed using ArcGIS in ESRI.

## 4. Results

### 4.1. Description of Soil Dataset

We described the SOC statistical information for the bare soil dataset and differentiated the selected models using calibration and validation datasets (Table 4). The mean value of SOC in the total soil dataset was 8.27 g/kg with values ranging from 4.495 to 17.284 g/kg. The coefficient of variation for the total dataset was 22.88%. When comparing the statistical characteristics of the calibration and validation SOC datasets, no significant differences were detected among the models (RF, SVM, PLSR, and ANN) or among the calibration and validation datasets for the mean, minimum, maximum, SD, and CV values. Because calibration and validation datasets were randomly selected for the four models, the kurtosis values were quite different among the models. The kurtosis was higher in the validation dataset (>0) than in the calibration dataset (>0) for all models except PLSR.

**Table 4.** General statistical description of soil organic carbon predictions from four models used in the western Guanzong Plain, China.

| Model | Samples | Mean g/kg | Min g/kg | Max g/kg | SD g/kg | Kurtosis | CV (%) |
|---|---|---|---|---|---|---|---|
| | Total | 8.27 | 4.495 | 17.284 | 1.897 | 1.581 | 22.88 |
| RF | Calibration | 8.236 | 4.495 | 14.152 | 1.816 | −0.077 | 22.04 |
| | Validation | 8.378 | 5.22 | 17.284 | 2.148 | 3.626 | 25.64 |
| SVM | Calibration | 8.22 | 4.495 | 14.152 | 1.861 | −0.154 | 22.642 |
| | Validation | 8.429 | 5.568 | 17.284 | 2.015 | 4.931 | 23.903 |
| PLSR | Calibration | 8.267 | 4.495 | 17.284 | 1.987 | 1.601 | 24.033 |
| | Validation | 8.281 | 4.698 | 13.05 | 1.595 | 0.123 | 19.257 |
| ANN | Calibration | 8.241 | 4.495 | 13.05 | 1.774 | −0.338 | 21.526 |
| | Validation | 8.362 | 4.698 | 17.284 | 2.258 | 3.083 | 27.002 |

RF, random forest; SVM, support vector machine; PLSR, partial least squares regression; ANN, artificial neural network; SD, standard deviation; CV, coefficient of variation.

### 4.2. Performance of the Prediction Models and Variable Importance

We built four SOC prediction models from S2A spectra (that included 10 single bands and 14 spectral indices) and from observed SOC from soil samples. The models were constructed by fitting the independent variable and measured SOC content through certain rules. During the process of model building, we found that not all variables contributed to model performance, so some of the independent variables were removed according to the VIP and RPD values. We ranked the importance of the variables in each model, then selected the top variables to input into the model, added one at a time, and iterated in turn until the change in RPD was no longer obvious. Finally, nine spectral bands and six selected spectral indices (shown in bold in Table 3) remained and were used as inputs into predictive models.

The prediction accuracy was quite different among the models, as evaluated by RMSE, $R^2$, and RPD (Table 5). In terms of RPD, the differences among the four models appeared to be significant; however, in terms of the other two indicators (RMSE and $R^2$), except for the PLSR model, the differences between the remaining three models were not so obvious, but should not be ignored. Generally speaking, the performance of the RF regression model was the best among all models, with an RPD value of greater than 2.25, $R_C^2$ of 0.873, and $RMSE_C$ less than 0.15. Therefore, the model has strong generalization and consistency, and its capability to quantitatively monitor SOC in the field is outstanding.

The ANN model also performed well, with RPD greater than 2.0, and $R_C^2$ greater than 0.73. However, its consistency and generalization were slightly weaker than the RF model. Compared with the ANN prediction model, all of the indices of the SVM model were lower ($R_C^2 = 0.6674$, $RMSE_C < 0.3$), but its RPD value was greater than 1.4, indicating intermediate performance of the SVM model. The predictive ability of the PLSR model was the weakest ($RMSE_C = 0.612$, $R_C^2 = 0.606$ and RPD = 1.3014), indicating that the stability and generalization of this model was too poor to be used for quantitative prediction (Table 5 and Figure 5).

**Table 5.** Model prediction performance statistics for random forest (RF), partial least squares regression (PLSR), support vector machine (SVM), and artificial neural network models applied to spectral image data from the western Guanzhong Plain, China.

| Model | Calibration Dataset | | Validation Dataset | | RPD | Formula |
|---|---|---|---|---|---|---|
| | $RMSE_C$ (g.kg$^{-1}$) | $R_C^2$ | $RMSE_V$ (g.kg$^{-1}$) | $R_V^2$ | | |
| RF | 0.146 | 0.8737 | 0.202 | 0.5712 | 2.2573 | y = 0.7468x + 2.025 |
| PLSR | 0.612 | 0.6060 | 0.265 | 0.5518 | 1.3014 | y = 0.6089x + 3.1471 |
| SVM | 0.208 | 0.6674 | 0.646 | 0.5967 | 1.4268 | y = 0.619x + 2.9723 |
| ANN | 0.196 | 0.7395 | 0.292 | 0.4251 | 2.0124 | y = 0.7857x + 1.7082 |

RF, random forest; SVM, support vector machine; PLSR, partial least squares regression; ANN, artificial neural network.

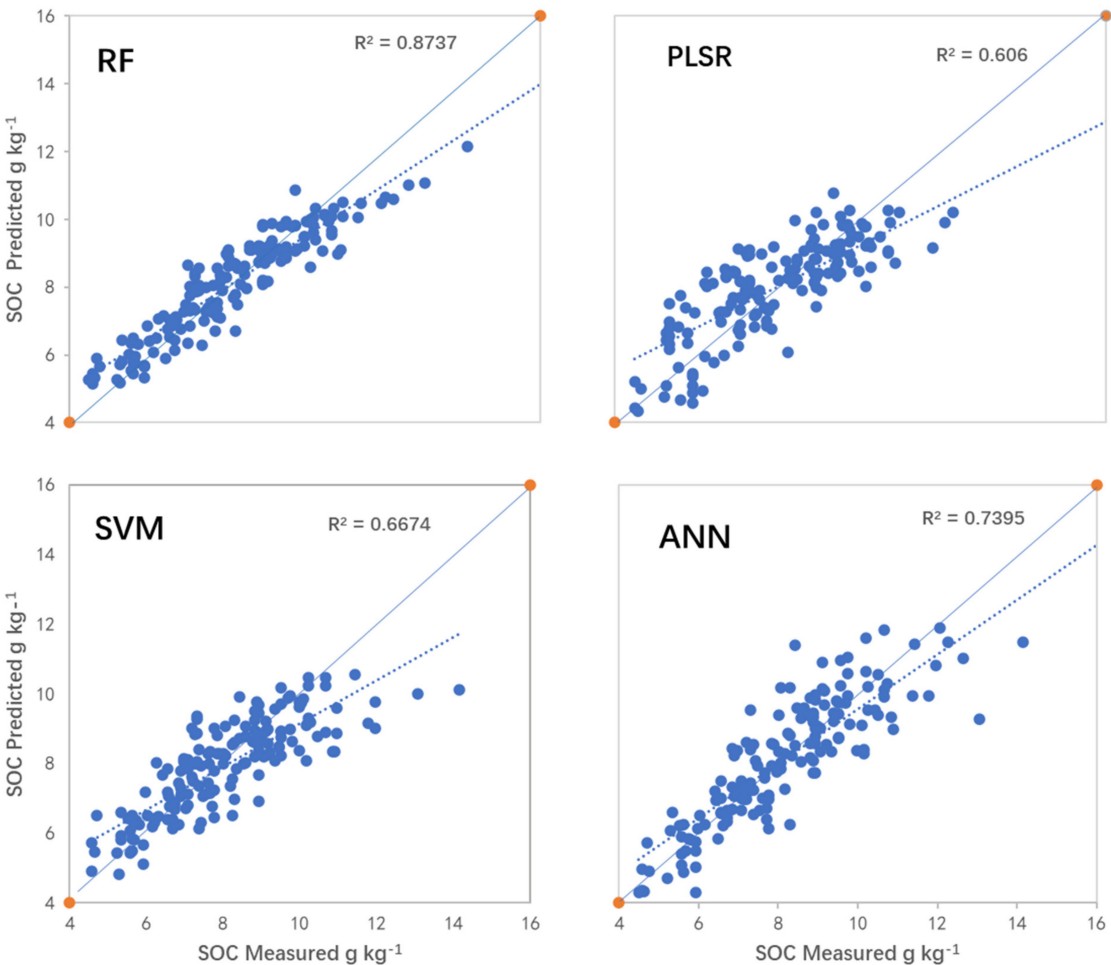

**Figure 5.** Predicted versus observed soil organic carbon (SOC) for the calibration dataset using four different models over the western Guanzhong Plain, China. RF = random forest; PLSR = partial least squares regression; SVM = support vector machine; ANN = artificial neural network.

The selected variables importance in models is shown in Figure 6. The figure shows that the variable with the highest relative importance in all models (except ANN) was B11, corresponding to the wavelength band of 1565–1655 nm, followed by B12 (1610 nm). This result is consistent with the higher reflectance of graded SOC content in the SWIR spectral ranges (Figure 2). It is significant that the variable importance of the RF and PLSR models had similar variation tendency, and the variables in SWIR have higher weight. These variables included B11, B12, and SATVI, while BI2 in the PLSR model was important as well. However, for the other two models, the variable importance in the visible wavebands was higher, especially in the ANN model. B6 had the highest weight in the ANN model, but the importance of variables in the SWIR should not be ignored.

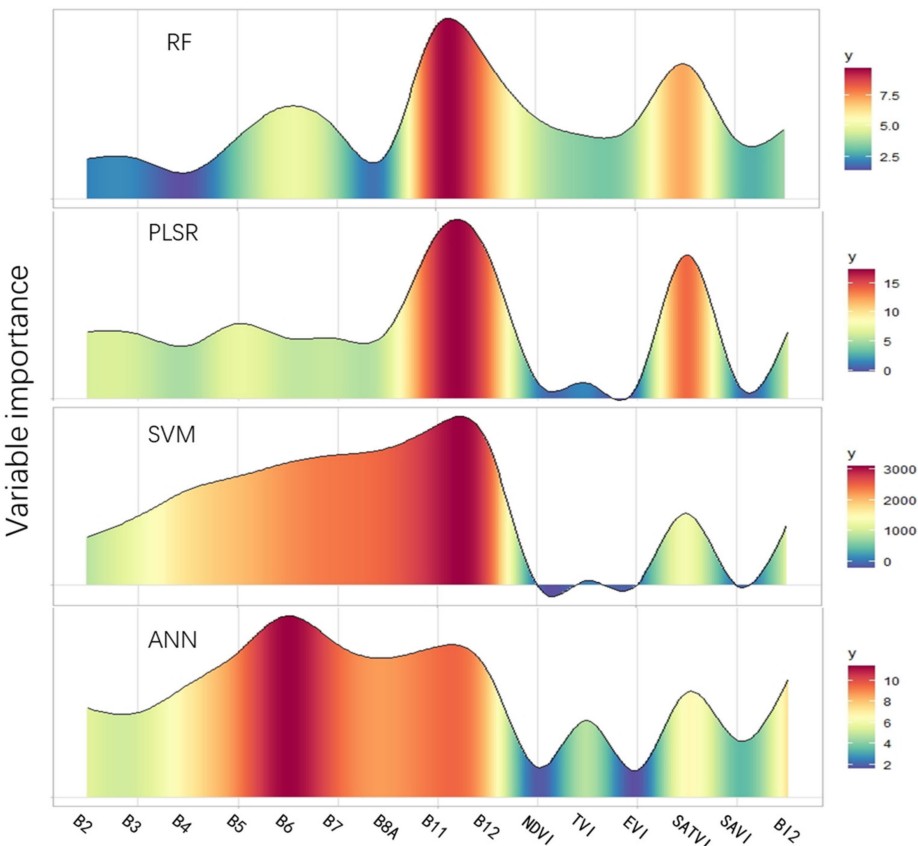

**Figure 6.** Distribution of variance importance in projection values for independent variables from four models used to predict soil organic carbon in the western Guanzong Plain, China. RF = random forest; PLSR = partial least squares regression; SVM = support vector machine; ANN = artificial neural network.

*4.3. SOC Map Variability Analysis*

Because the RF model had the best prediction accuracy, it was applied to retrieve the SOC distribution of all bare cropland in the study area. The resultant dataset was used to map SOC (Figure 6) in order to quantify, simulate, and interpret the dependence of SOC on the spatial environment of the study area through Kriging technique. Because of the left-skewed distribution, a BOX-COX transformation was carried out. In the process of interpolation, it was found that SOC had no anisotropy in the north-south and east-west directions. Therefore, an isotropic nested model was used to fit the environmental variance (Figure 7). The semi-variance fitting model contains two basic structures, the nugget effect and the exponential model. The nugget value was 1.33597, and the partial sill, i.e., the difference between sill and nugget values, was 0.8295. The results showed that spatial dependence of SOC in the study area was marginally obvious, even variability was

calculated when the distance was less than 1500 m, and the maximum semi-variogram is only 0.021.

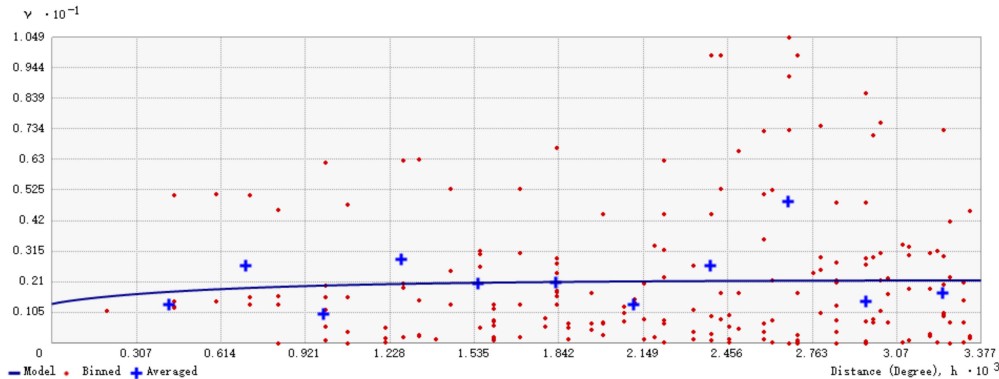

**Figure 7.** Semi-variograms computed for soil organic carbon from the predicted dataset of bare soils in the western Guanzhong Plain, China.

In order to better express the geographical differences of the SOC content, the interpolation results were manually divided into five classes and shown on a map of the Guanzhong Plain study area (Figure 8) in which the spatial dependence of SOC is shown. In the lower terrace of the Wei River, the SOC content was generally higher, and about 60% of the fifth class of SOC (9–12.7 g/kg) was found here. The SOC content of the higher terrace and Loess Plateau far away from the Wei River was relatively low. From the perspective of administrative divisions, the soil of the Chencang district has relatively high SOC content, followed by Qishan county because of proximity to the Wei River. Moreover, SOC content over the study area ranged from 5.39 g/kg to 12.7 g/kg, with an average value of 7.23 g/kg. The interpolation result was still a left-skewed distribution, with more than two-thirds of the area belonging to 5–8 g/kg class.

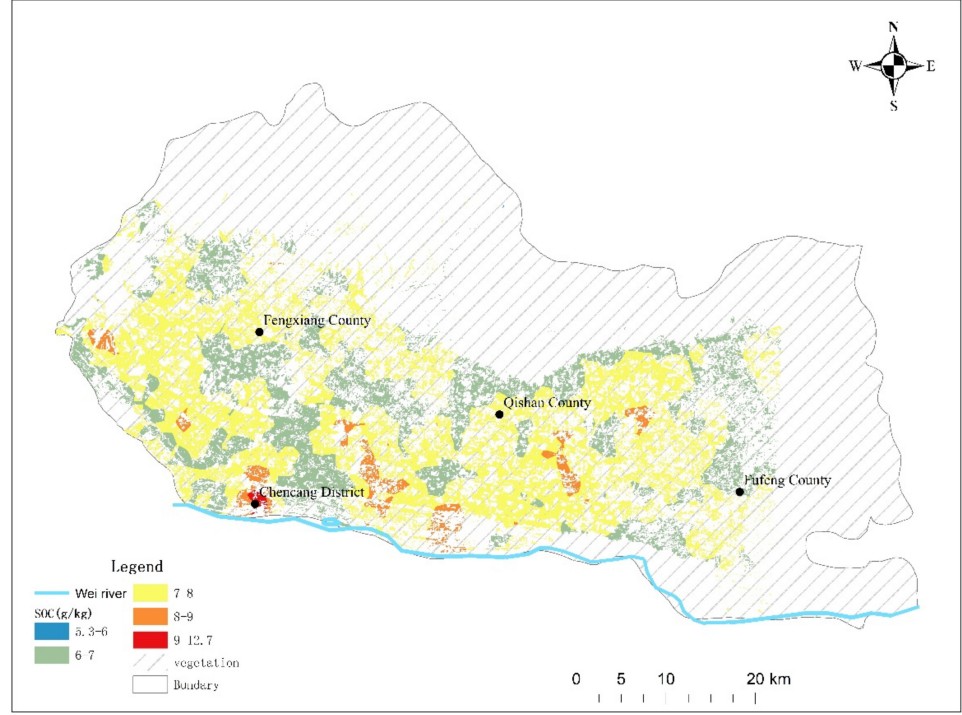

**Figure 8.** Predicted soil organic carbon (SOC) spatial distribution in the western Guanzhong Plain, China, based on the random forest model.

## 5. Discussion

Rapid progress in satellite sensors and easy access to remote sensing images has aroused great attention for research on soil classification and soil property prediction based on spectral reflectance information, resulting in the promotion of comprehensive exploration of soil digital mapping and soil remote sensing. The S2A satellite provides a new generation of multispectral imagers equipped to acquire some wavelengths in the SWIR region, and is considered to be an intensely interesting remote sensing data source related to the reflectivity features of SOC. Both Vaudour and Castaldi et al. evaluated the capability of S2A data for soil property prediction, and reported positive potential [25,26]. However, only intermediate or near intermediate prediction performance outcomes were obtained, with low $R^2$ (0.3–0.5) and RPD (<1.4) for seven soil attributes in results from Vaudour et al. [26]. Castaldi et al. [25] found that only three out of seven study areas obtained satisfactory prediction accuracy (RPD > 2) when evaluating the capability of the S2A data for SOC prediction in croplands. The non-negligible factor that determines soil attribute prediction accuracy based on spectral reflectance is the vegetation on the soil surface. Gholizadeh et al. [24] selected four areas less than 1 km$^2$ in central Europe for SOC inversion and verification. They reported moderate model performance (maximum RPD was 1.92). During our field investigation, we found a large area of continuous bare land that was not planted in the autumn and had an area of 773.44 km$^2$, accounting for 26.43% of the total area of the western Guanzhong Plain. This area of autumnal bare land provided the opportunity to eliminate the effects of vegetation on soil property prediction based on S2A data. We found that we could accurately predict SOC with higher $R^2$ (ranging 0.6674 to 0.8737) and RPD (the highest value was 2.2573) than reported in previous research results.

Selecting the correct prediction model was an essential factor determining soil attributes prediction accuracy from multispectral images. Based on the spectral reflectance values extracted from the same satellite image and the same sample dataset, the current research compared the ability of training soil surface SOC and predicting SOC by RF, PLS, SVM, and ANN regression algorithms. Even though Castaldi et al. [25] indicated that the performance of the RF model was not optimal and that the correlation between variables was weak, the RF algorithm was still used to bond reflectance and SOC, and a predictive model with high consistency was, indeed, obtained (Table 5 and Figure 5). Even though PLSR is the most commonly used multivariate statistical technology in soil science [65–68], it was not satisfactory in this study. This may be because PLSR is a linear model, but the relationship between organic carbon and the soil spectrum may be nonlinear, especially for the region with strong variability and large range. While RF is a nonlinear model, it performed regression tasks by multiple decision regression trees, and retrieved the surface organic carbon content value by weighted average of regression results according to variable importance (Figure 6). Meanwhile, RF makes comprehensive and in-depth data mining analysis on independent variables both in regression and classification [69–71]. Therefore, RF had more advantages than PLSR in this study. In addition, model performance is not only related to the algorithm itself. The tendency of variable importance in the ANN model was significantly different from the other models. This may be because ANN is an unsupervised algorithm. Farifteh et al. [35] reported that both methods have a great potential for retrieving and mapping soil salinity, but PLSR performed better, after comparing PLSR and ANN in quantitative analysis of salt affected soil reflectance spectra, because the relation between soil salinity and soil reflectance can be approximated as a linear correlation. Although the importance of variables in the SWIR was high for all of the models, there were obvious differences in the importance distribution of other variables. It was also shown that, in line with previously reported results, S2A has outstanding inversion ability for soil properties [23], and may be higher than the Landsat series, the SPOT series, and other land resource satellites. Therefore, S2A is expected to be applied at a large scale in the future.

In recent years, many scholars have explored the relationship and differences between soil properties and spectral reflectance, as well as between soil properties, including

SOC, soil texture (sandy soil, clayey soil, and loam), CEC, pH, CaCO$_3$, and iron content, among which SOC and soil texture have been more widely studied [72]. If there is a direct correlation between attributes, then the spectral sensitive attributes can be used to predict attributes not sensitive to the spectrum. In fact, both pH and CEC are insensitive to the spectrum. Gholizadeh et al. [24] found a relationship between soil texture and SOC. Relatively high organic carbon content is more likely to be found in areas with higher clay and silt content, especially clay content, because with the increase of clay and silt content, soil aggregate or agglomeration capability will be enhanced, thereby avoiding the rapid or direct degradation of organic carbon. SOC concentration in sandy soils has a positive influence on the CEC, and higher pH is often associated with higher SOC in sandy soils [73]. However, Goidts et al. [74] studied the change of SOC on cultivated land surface over a 50-year period, and found that soil texture was not one of the significant driving factors of SOC change. In this study, large SOC content ranges (4.495–17.284 g/kg, Table 4) and negative correlations between SOC and S2A spectral reflectance were detected (Figure 3), indicating that SOC is a vital factor affecting spectral reflectance characteristics. Soil moisture concentration influence soil reflectance (i.e., soil reflectance decreases with increasing soil moisture [28]). Soil texture affects soil spectral reflectance by affecting soil water storage capacity, resulting in differences in soil water content. Soil texture also affects soil particle size, i.e., soil roughness, and thereby has a significant impact on soil reflectance [75,76]. The western Guanzhong Plain is located in the Wei River valley and is surrounded by the Qinling Mountains and the Loess Plateau in the south, west and north direction. Therefore, the climate and soil were relatively homogeneous with similar precipitation and evaporation, and loamy soil texture. Therefore, the soil moisture and soil texture were not included in the SOC prediction models in the current research [37].

In fact, hyperspectral datasets obtained in the laboratory have a strong predictive ability for the soil properties mentioned in this paper. SOC and clay are more sensitive to the spectrum, so they are widely used and mostly meet expectations. However, the performance of inversion models for the same attribute and evaluated by the same indicators in different studies is not always the same, likely because sample selection methods, measuring instruments, sample processing, spectrum acquisition methods, and analysis algorithms all play important roles [10,77]. Until now, multispectral data have been mainly used for soil classification in digital thematic maps. However, with the continuous improvement and optimization of radiation resolution, spatial resolution, and band location of multispectral sensors, multispectral data now has the ability to retrieve soil property information. In particular, the new generation of sensors can simultaneously obtain the reflectance spectrum information of visual-near infrared (VNIR) and SWIR wavelengths, covering the range of typical spectral characteristics of soil. Gholizadeh et al. [24] compared the retrieval and mapping capabilities of the S2A, airborne, and near-end handheld sensors for SOC and soil texture. The results showed that the three types of spectral data were all suitable for SOC monitoring in most areas, but the performance of the clay content retrieval model was slightly weaker. At the same time, with the gradual increase in the platforms' height, the ability of a sensor to predict soil parameters gradually decreases, especially with regard to soil particles and sand particles that are less sensitive to the spectrum. After all, the light source, signal-to-noise ratio, spatial resolution, atmospheric conditions, and pixel purity are quite different among the three platforms. In addition, the actual remote sensing product can be disturbed by many factors, such as soil roughness, soil moisture, and soil cover. Even so, some studies on soil property monitoring using S2A as a single data source have overcome many problems and have achieved certain results, and the verification results are consistent with the conclusions of this study, i.e., S2A has the capability to monitor soil properties, and it is expected to develop the potential for retrieving soil properties.

It should be noted that there are still some limitations in our research. First, space-borne data are controlled by the data acquisition condition, which mainly associates with atmospheric attenuation and plant litter, and the acquisition time must be as close as

possible to the sampling time. Second, we did not take into account the effect of the soil moisture and soil texture on SOC prediction because they were assumed to be relatively homogeneous by us, but this needs to be confirmed. Third, the simultaneous existence of large bare tilled agricultural fields and high-quality remote sensing images is an important prerequisite for replicability.

In the future, the continuous research of machine learning and deep learning will promote further development of remote sensing inversion technology. On the one hand, more algorithms will be introduced to retrieve land surface parameters. These algorithms include genetic algorithm and multivariate adaptive regression splines functions that have been applied in recent years. On the other hand, the traditional algorithms (RF, PLSR, etc.) will be analyzed and integrated to better assimilate the land surface process model and remote sensing data. In addition, even with continuous sensor improvement and increasing data quality, it is difficult to monitor soil properties and conduct mapping using a single image or sensor on a large scale, let alone implement the direct application of the results by end users. Therefore, it is necessary to incorporate S2A data into the large framework of digital soil mapping. Interested users may use variograms that closely resemble those obtained from actual measurements as proxy input data to quantitatively analyze specific soil properties, or to directly optimize digital soil mapping models.

## 6. Conclusions

We studied the correlation between the bare soil spectrum and soil surface organic carbon in the western Guanzhong Plain of China. We tested the ability of S2A data to retrieve SOC content in bare soil. The performance abilities of four machine learning models to link spectral reflectance data and measured SOC content were compared. The results showed that the RF model had the highest prediction accuracy, followed by the ANN model. Regarding the importance of prediction variables in the models, B11 and B12 were of high importance in all four models, and SATVI performed unstably in the models. The SOC map derived from the RF model showed that the spatial resolution and spectral resolution of S2A were good enough to monitor SOC content at the regional scale, and to describe the spatial dependence of soil structure. In addition, in order to carry out large-scale organic carbon monitoring, having bare land without obvious foreign matter (vegetation) coverage is very important, and the terrain, soil moisture, and soil quality of the study area will also play an important role. Future studies may be necessary to continue to verify the monitoring performance of S2A spectral data with regard to soil properties in different environments, to explain the dependence of soil properties on structure, and to use more auxiliary data or auxiliary data from digital models to improve model prediction accuracy. In recent decades, more sensors with high spectral collection efficiency in the SWIR (e.g., EnMAP, PRISMA, HyspIRI) will greatly contribute to the study of spectral characteristics of soil properties, such as clay and calcium carbonate content.

**Author Contributions:** Conceptualization, K.W. and W.G.; methodology, J.Z.; software, K.W.; validation, K.W., W.G. and J.Z.; formal analysis, K.W.; investigation, K.W.; resources, Y.Q.; data curation, K.W.; writing—original draft preparation, K.W.; writing—review and editing, Y.Q.; visualization, K.W.; supervision, Q.C.; project administration, Y.Q.; funding acquisition, Y.Q. All authors have read and agreed to the published version of the manuscript.

**Funding:** This research was funded by the National Natural Foundation of China, 41877007 and the National Training Program of Innovation and Entrepreneurship for Undergraduate, S202010712331 and S202010712104; the APC was funded by 41877007.

**Conflicts of Interest:** The authors declare no conflict of interest.

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
