# Peer review of "Retrieval and Mapping of Soil Organic Carbon Using Sentinel-2A Spectral Images from Bare Cropland in Autumn"

_remotesensing, doi:10.3390/rs13061072_

Round 1
Reviewer 1 Report
The manuscript develops a model for predicting SOC in cropland. However, there are many studies currently focused on the application of different Remote sensing platforms for predicting SOC in croplands. The most recent published article is https://doi.org/10.3390/rs13020308. Therefore, this manuscript doesn't have any novelty. On other hand, the developed model is for a new region.
Although authors discussed the relationship between SOC and other soil properties in other studies, they have not discussed their related findings. I suggest adding it to their discussion.
Table 3: what the question marks in the table?
Figure 2: The flow chart is not correct, authors are not using all algorithms together in order to model SOC.
Reviewer 2 Report
The title of the manuscript (MS) deals with "Retrieval and mapping of soil organic carbon using Sentinel-2A spectral images from bare cropland in autumn". The topic of this manuscript is of interest and well written and I liked reading it, great job!
Just one comment.
In the "discussion" section, the author must extend the comparison between their approach and other ones that have been developed and used in the literature for the same or related purposes (I recommend increasing the number of Scientific articles cited, especially to compare the study context with similar studies). Also, in this section, the authors should also highlight the current limitations and usefulness of the proposed research, and briefly mention some precise directions that they intend to follow in their future research work.
Reviewer 3 Report
The authors presented an interesting study on SOC estimation through S2A data highlighting the potential and limits of this application. Their field data are adequate and widespread with good variability.
From my point of view, the main issues are with
- the presentation of the Materials section that is confused. This section needs to be accurately rewritten, clearly presented, and described;
- some important missing points in the discussions and conclusions
Lines 78-80: S2A is a multispectral sensor with 13 bands and here you call it as super-spectral and hyperspectral. I suggest calling S2A data as multispectral in the entire manuscript. Revise it.
Line 157: what do you mean with “accumulated temperature above 10 ℃ is about 4500℃.”
Section 2.2. "Super-spectral satellite data pre-processing and retrieval of indices" is not a pertinent title for this subsection. I would rather remove this section and move these sentences (Lines 170-184) before a new section 2.2 that includes also the actual section 2.3. The title of the new section 2.2. could be "Soil sampling and S2A data".
Lines 160-163: add a reference
Line 184: an is a
Line 187: super-spectral change into multispectral
Line 207: please justify why you selected only 9 of the 13 bands a priori.
Lines 252-253: show the results of the VIP for each band for the SOC and which threshold you chose for their exclusion.
3.2 title: typo error
Lines 448-449: add here in the discussion the other factors affecting soil attribute prediction accuracy in particular SOC. For example, soil roughness, soil moisture, soil texture, and so on. Add a proper description of these factors in this section with the relative references. Soil moisture strongly influences the soil spectral behavior and also the estimation of SOC! The authors did not take into account these limitations in their study. It is therefore noteworthy that you write a consistent description of it in this section.
Line 553: the title should be Conclusions
In the conclusions and also in the discussions and introduction, you should add the main limitation of this procedure that is strictly related both to the only one short period of S2A acquisition (3 near dates) and to the best possible "conditions" you chose for retrieving SOC of the first 30 cm of agricultural soils, i.e., bare soil (I assume you mean tilled agricultural fields). Moreover, you don’t take into account the soil moisture of you soils samples which is another great limitation in its replicability. These limitations should be added in the entire manuscript!
Round 2
Reviewer 1 Report
The authors revised their manuscript according to my comments and improved it. Still, I didn't see any novelty in the manuscript except the new study area; however, the authors revised their discussion so that the manuscript became compelling.
There are many sentences in the manuscript borrowed from published articles, but it is not cited. For instance, authors in lines 243-247 borrowed it from Gholizadeh et al. 2018 [doi:10.1016/j.rse.2018.09.015] without citing it here. I highly encourage authors to avoid it and check and revise their whole manuscript accordingly.
Reviewer 3 Report
The authors have substantially improved the manuscript following the reviewers' comments.
Minor comments:
Line 182: put the new reference in the standard format and update the references.
Line 587: add some examples, e.g., PRISMA (ASI Italy) hyperspectral data
